# Profile of Back Bacon Produced From the Common Warthog

**DOI:** 10.3390/foods9050641

**Published:** 2020-05-15

**Authors:** Louwrens C. Hoffman, Monlee Rudman, Alison J. Leslie

**Affiliations:** 1Department of Animal Sciences, Faculty AgriSciences, Mike de Vries Building, Private Bag X1, Matieland, Stellenbosch University, Stellenbosch 7602, South Africa; monlee.swanepoel@gmail.com; 2Centre for Nutrition and Food Sciences, Queensland Alliance for Agriculture and Food Innovation, The University of Queensland, Coopers Plains, QLD 4108, Australia; 3Department of Conservation Ecology and Entomology, Faculty AgriSciences, JS Marais Building, Private Bag X1, Matieland, Stellenbosch University, Stellenbosch 7602, South Africa; aleslie@sun.ac.za

**Keywords:** warthog, game meat, *Phacochoerus africanus*, bacon, sensory, flavor, processed products

## Abstract

The common warthog (*Phacochoerus africanus*) has historically been hunted and consumed by rural communities throughout its distribution range in Africa. This study aims to develop a processed product from warthog meat in the form of back bacon (*Longissimus thoracis et lumborum*) as a healthy alternative meat product and to determine its chemical and sensory characteristics derived from adult and juvenile boars and sows. The highest scored attributes included typical bacon and smoky aroma and flavor, and salty flavor, as well as tenderness and juiciness. Neither sex nor age influenced the bacon’s chemical composition; the bacon was high in protein (~29%) and low in total fat (<2%). Palmitic (C16:0), stearic (C18:0), linoleic (C18:2ω6), oleic (C18:1ω9c), and arachidonic (C20:4ω6) were the dominant fatty acids. There was an interaction between sex and age for the PUFA:SFA ratio (*p* = 0.01). The differences between sex and age class are considered negligible regarding the overall profile and healthiness of the bacon.

## 1. Introduction

The global trend toward healthier and sustainable living promotes the increased production and consumption of the meat from wild animals, also referred to as game meat or venison. Game meat production is promoted as a low-input alternative to traditional animal husbandry [1,2,3]. In general, game animals produce a lean and healthy meat, which is high in protein, low in fat with a favorable fatty acid profile. The low overall lipid content and fatty acid composition is largely attributed to their forage diet and high levels of activity [4]. Most consumers show a preference for meat and meat products with low amounts of visible fat, and are willing to pay more if they are actually lower in fat as it is associated with increased quality [5].

Although the formal game meat industry produces sizable quantities of game meat in many southern African countries annually [6,7], it has been suggested that general ignorance regarding the quality aspects of game meat and preparation methods has been crippling the growth of the fresh game meat industry [8]. Meat consumption is primarily influenced by availability, price, and tradition [9]. It is also possible that consumers associate game meat with lower quality and flavor attributes and therefore desirability, while they might also be unfamiliar with game meat species [8,10]. However, despite the relative slow growth of the game meat industry, consumers and producers have increasingly become aware of the health attributes of game meat in general and its value as a sustainable red meat source [11]. Game meat is also considered exotic, which is attractive to consumers who are adventurous and to foreign tourists to South Africa who want to consume meat from native animals [10,12,13].

The meat from wild animals has been associated with a gamey or livery taste [14]. In addition, game meat and the meat from non-ruminants is typically high in total unsaturated FA (UFA) content and thus more susceptible to lipid oxidation [15,16]. Developing processed game meat products is a potential strategy to introduce the meat of different game species to the commercial market, as the addition of preservatives and anti-oxidants, together with smoking and curing is used to inhibit and/or mask lipid and protein oxidation, thereby increasing shelf life and color stability while imparting specific flavors [17,18,19,20]. Consumers also appreciate meat products that resemble traditional products in appearance and eating sensation [21].

A recent report by the World Health Organization’s (WHO) International Agency for Research on Cancer (IARC) stated that the consumption of red meat and processed meat products is associated with the risk of certain types of cancer. The IARC placed processed products in the same category as asbestos, second-hand tobacco smoke, and gamma radiation, claiming that consuming >50 g of processed meat daily increases the risk of colorectal cancer by 18%. The study recommends reduced intake of red meat and processed products but acknowledged that meat is still a valuable source of high nutritional value [22]. Therefore, future meat product development should aim to meet global health recommendations in terms of low total fat and saturated (SFA) content, with favorable polyunsaturated to saturated fatty acid (PUFA:SFA) and omega 6 to omega 3 (n-6:n-3) ratios and decreased preservative (such as salt) content. Game meat has successfully been used in a variety of processed products with beneficial health benefits. For example, studies found that meat products produced from game meat in comparison with domestic animal meat have lower fat contents and higher nutritional values [23]. Considering the popularity of game meat products such as biltong in South Africa, and alheira sausages in Portugal, increasing research into game meat processing broadens the scope of wild animal production/consumption and global meat provision [24]. Warthog meat has been successfully used in the production of cabanossi [25] and when combined with ostrich meat and olive oil, was found to be an acceptable health product [26].

The common warthog (*Phacochoerus africanus*) has historically been hunted and consumed by rural communities throughout its large distribution range across Africa. Currently the species also occurs extra-limitally in parts of South Africa through deliberate introductions and subsequent range expansion. Warthogs are hunted by agricultural producers for damage reprisal, and by recreational and trophy hunters [27]. This produces a carcass of which the meat can be used for human consumption, as warthogs have a high dressing percentage, low total intramuscular lipid content, high total protein and moisture content, and a favorable fatty acid profile [25,28,29]. This study aimed to develop a processed product from warthog meat in the form of back bacon (*Longissimus thoracis et lumborum*) as a healthy alternative meat product, and to determine the bacon’s chemical and sensory characteristics.

## 2. Materials and Methods

### 2.1. Sample Collection and Preparation

Free-roaming warthogs were culled on the Pongola Game Reserve (27°22′09.26″ S, 31°50′42.16″ E) situated in KwaZulu Natal province in South Africa. A total number of 31 warthogs were shot during the daylight hours using single shot bolt action rifles (Ethical clearance number: 11LV_HOF02) when sighted, with bias toward sex and age until the desired quota per group (*N* = 8) was achieved. Unfortunately, only seven warthogs for the group of juvenile males were culled as the full quota could not be obtained within the given time period. Although other studies used tusk protrusion and molar eruption as the basis for classifying age classes [30], the data capturing team had difficulty in judging their age in the field during culling as warthogs have a secretive and avoidance behavior toward humans/vehicles, animals were shot as they were observed with a certain bias toward sex and age, as it was not desired to cull animals unnecessarily. Therefore, tusk protrusion and estimated body weight were used as a visual tool to determine age classing. In support of this decision, the animals weighing less than 35 kg, with no tusk protrusion past the flanges of the lips were classified as juveniles, and all other animals classified as mature adults.

Immediately after shooting, the animals were exsanguinated by thoracic sticking, transported to a slaughter facility, and weighed and dressed according to the *Guidelines for the Harvesting and Processing of Wild Game in Namibia* [31]. The body weight of the animal was defined as the total weight of the animal (minus the blood lost during exsanguination). Dressing entailed the removal of the head, feet, and skin. The head was removed by making a horizontal cut between the axis and atlas bones of the neck vertebrae. The trotters were removed by making a horizontal cut through the metacarpal carpal joint (joints between the carpal joints and radius and ulna). The skin was removed by starting at the anus and working toward the neck area, while attempting to leave as much subcutaneous fat on the carcass as possible. After evisceration the carcass was washed, allowed to drip dry for ±20 min before being weighed again to determine hot carcass weight. After the carcass was stored at 4 °C it was weighed again to determine cold carcass weight (24 h post mortem). The right *Longissimus thoracis et lumborum* (LTL) muscles were excised from the chilled carcass and weighed, vacuum packed, and frozen at −20 °C before being transported to Stellenbosch University. The LTL was used for bacon production and subsequent sensory and chemical analyses.

### 2.2. pH

A pH measurement was taken from the center of the LTL muscle (~3rd last rib) ±45 min after exsanguination using a Crison pH25 handheld portable pH meter (Lasec (Pty) Ltd., Cape Town, South Africa). Before each reading, the meter was calibrated with standard buffers (pH 4.0 and pH 7.0) as provided by the manufacturer. A final measurement was at taken 24 h after the LTL muscles were excised and considered as ultimate pH (pH_u_).

### 2.3. Bacon Production

The recipe and spices (Ready Brine^TM^) for the bacon was sourced from a commercial producer, Deli Spices (25 Bertie Avenue, Epping 2, Cape Town, South Africa) and to 10 L of water, 1.5 kg of brine [Ref: 01124015] is added (Table 1). The LTL muscles were thawed overnight (12 h) at 4 °C. The thawed muscles were dabbed dry with a paper towel and weighed (loin weight) to determine thaw loss. It was initially envisaged to produce back bacon following commercial processing methods (i.e., by injection) to achieve a pick-up of 20%, but since the muscles were small and differed in size between ages it was decided to submerge the muscles in a brine solution for ±72 h, at fridge temperatures of 3.9 °C (±standard deviation [SD] 0.37). After brining, the LTL were weighed (to calculate “final pick-up %” relative to loin weight) before being smoked using English oak blocks in a commercial smoker Reich Airmaster^®^ UKF 2000 BE (Reich Klima-Räuchertechnik, Urbach, Germany) with a SmartSmoker and TradiSmoker LS 500 HP electronic, automatically controlled by a Microprocessor (Unicontrol 2000), according to the program as set out in Table 2. The relative humidity was controlled by the chamber. A thermocouple probe was inserted into one medium sized bacon sample to measure the change in core temperature during the smoking program, while the chamber provided the measurements of chamber temperature and chamber relative humidity. The changes are illustrated in Figure 1. After smoking, the LTL were weighed again to determine the “smoking loss %” relative to the “loin weight” for calculation of total weight change (%), vacuum packed, and frozen at −20 °C until further analysis.

### 2.4. Descriptive Sensory Analysis.

The entire LTL bacon was divided to roughly represent the *Longissimus thoracis* (LT) (T_3_ to T_12_/T_13_) and *Longissimus lumborum* (LL) (T_12_/T_13_ to L_5_) portion, with the separate portions being weighed again. The visible fat and tendons were removed before the LT portion was homogenized, vacuum packed, and stored at −20 °C for chemical analysis (proximate and fatty acids). The LL portion was vacuum packed and stored at 4 °C prior to sensory analysis the subsequent day. The LL portion for sensory analysis was removed from vacuum package, gently blotted dry with a paper towel, and the whole (i.e. not sliced) LL was placed in individually marked oven bags (Glad^®^), and prepared according to Geldenhuys and co-workers [32]. After cooking the LL was removed from the bags and allowed to cool for 10 min before being weighed to determine cooking loss, and subsequently cut into 1.2 cm^3^ cubes. The cubes were wrapped in aluminum foil and placed in ramekins and reheated at 100 °C for 7 min. After preheating the meat was covered with Petri-dish lids and placed on half-filled cups in water-baths heated to 70 °C in order to maintain temperature.

A panel of 10 judges were trained using standard, generic descriptive techniques [33]. The panelists were recruited from a pre-existing group of judges involved with research studies on meat at Stellenbosch University. During six one hour long training sessions the panelists made use of reference samples to formulate a list of sensory attributes, including aroma, flavor, and texture. The sensory attributes and reference samples are described in Table 3, with most of the reference samples being toward the upper end of the scale. The cooked bacon samples (LL) were evaluated by the trained panelists during seven blind-tasting sessions of ±45 min that scored each attribute on an unstructured 100-point scale. The cooked samples were presented to, and the data collected from, panelists using Compusense^®^ five software (Compusense, Guelph, Canada) while seated in booths in a temperature (21 °C) and light (artificial daylight) controlled room. Each panelist received three samples of warthog bacon randomly assigned to the session per treatment. The test reproducibility of the trained panel was determined using test-retest [32].

### 2.5. Proximate Analyses

The total moisture, ash, and crude protein content (%) were determined on the LT back bacon according to the Association of Official Analytical Chemist’s Standard Techniques (AOAC) method 934.01 [34] (for moisture content), method 942.05 (for ash content) [35] and the Dumas combustion method 992.15 (for crude protein content) [36]. Total moisture content was determined using a 2.5 g homogenized sample. The moisture-free sample was used to determine the total ash content. A 0.15 g, defatted, dried, and finely ground sample was used for crude protein analysis with a Leco Nitrogen/Protein Analyzer (FP-528, Leco Corporation, Saint Joseph, MI, USA). Before the analyses the machine was calibrated using 0.15 g ethylenediamineteraacetic acid (EDTA) samples (Leco Corporation, 3000 Lakeview Avenue, St. Joseph, MI 49085-2396, USA, Part no.502-092, Lot no. 1055). After 20–30 analyses the machine was again calibrated with EDTA samples. The percentage nitrogen (% N) per sample was determined and multiplied by a conversion factor of 6.25 to calculate the total crude protein content per sample. Chloroform/methanol (1:2, *v*/*v*) was used to determine the total lipid content using a 5 g sample [37]. The laboratory at the Department of Animal Sciences, Stellenbosch University, is accredited by the Agricultural Laboratory Association of South Africa (AgriLASA) to perform accurate and reliable proximate analyses and for validation of accuracy and repeatability, partakes in monthly inter-laboratory blind tests.

### 2.6. Shear Force

The Warner Bratzler shear force (WBSF) test was used to determine the instrumental shear force of the cooked back bacon (LL) samples [38]. The samples were wrapped in aluminum foil, placed in plastic bags and refrigerated at 4 °C for three days (72 h) before being subjected to analyses. Three adjacent 1 cm × 1 cm meat strips were cut parallel to the muscle fiber direction, and cut further into a minimum of six rectangular cubes (1 × 1 × 2 cm). An Instron Universal Testing Machine attached with a WBSF fitting was used to determine the shear force required to cut the meat sample perpendicularly to the muscle fibers at a crosshead speed of 200 mm/min. Shear values were recorded in Newton (N) and an average was calculated according to the number of samples.

### 2.7. Fatty Acid Analysis

The fatty acids were extracted from a 2 g homogenized LT bacon sample using chloroform/methanol (1:2, *v*/*v*) solution [39]. The extraction solution contained 0.01% butylated hydroxytolene (BHT) to act as an anti-oxidant. A 2 g meat sample and 20 mL solution was homogenized using a polytron mixer (Kinematica, type PT 10-35, Switzerland). Heptadecanoic acid (C17:0) was added (0.5 mL) to the homogenized sample as an internal standard in order to quantify the observed fatty acids in the sample (Internal standard: Catalogue number H3500, Sigma-Aldrich Inc., 3050 Spruce Street, St. Louis, MO 63103, USA). A 250 µL sub-sample of the extraction was transmethylated for 2 h at 70 °C in a water-bath using methanol/sulphuric acid (19:1, *v*/*v*) as transmethylating agent (2 mL). The transmethylated sample was cooled to room temperature before the fatty acid methyl esters (FAME) were extracted by adding 1 mL dH_2_O and 2 mL hexane to the sample and transferring the top hexane layer. The sample was dried under nitrogen, after which 50 µL hexane was added, of which 1 μL was injected into the gas chromatograph. The FAMEs were analyzed using a Thermo TRACE 1300 series gas-chromatograph (Thermo Electron Corporation, Milan, Italy) equipped with a flame-ionization detector, using a 30 m TR-FAME capillary column with an internal diameter of 0.25 mm and a 0.25 µm film (Cat. No. HY260M142P, Anatech, Cape Town, South Africa) and a run time of ca. 40 min. The following oven temperature settings were utilized: Initial temperature of 50 °C (maintained for 1 min) and final temperature of 240 °C attained after three ramps (initial increase at a rate of 25 °C/min until a temperature of 175 °C was reached; thereafter an immediate increase at a rate of 1.5 °C/min to reach 200 °C and maintenance of this temperature for 6 min; lastly an increase at a rate of 10 °C/min to reach 240 °C and maintenance of this temperature for a minimum of 2 min). The injector temperature was set at 240 °C and the detector temperature at 250 °C. The hydrogen gas flow rate was 40 mL/min. The FAME of each sample was identified by comparing the retention times with those of a standard FAME mixture (Supelco™ 37 Component FAME mix, Cat no. 47885-U, Supelco, Bellefonte, PA, USA), with results being expressed as mg fatty acid/g meat. Results were given as a milligram per gram of fatty acids present in bacon.

### 2.8. Statistical Analysis

The study was a completely random factorial design with seven warthogs harvested at random for each of two age classes (juveniles and mature adults) and two sexes (males and females). Randomly selected warthog back bacon samples from each of the four treatment combinations (two age classes × two sexes) were assigned to each of the seven descriptive sensory analysis sessions. During the testing phase, the performance of the sensory panel was monitored using the software program Panelcheck (Version 1.3.2, www.panelcheck.com). Panel reliability was finally tested by subjecting the data to an analysis of variance (ANOVA) model similar to 5.7 in Næs et al. [40] using the GLM (General Linear Models) procedure of SAS™ software (Statistical Analysis System, Version, 9.2, 2006, SAS Institute Inc., Cary, NC, USA). The model is indicated by the following equation:*y_ijk_ = µ + a_i_ + r_j_ + p_k_ + ar_ij_ + ap_ik_ + rp_jk_ + ε_ijk_*(1)
where *a_i_* is the assessor effect, *r_j_* is the session effect, *p_k_* is the product effect, *ar_ij_, ap_ik_,* and *rp_jk_* are the interaction effects, and *ε_ijk_* is the random replicate error. Further pre-processing of the sensory and physical data involved performing Shapiro–Wilk tests on the standardized residuals from the model to test for deviation from normality [41]. In cases where there was significant (*p* ≤ 0.05) deviation from normality outliers were removed when the standardized residual for an observation deviated with more than three standard deviations from the model value.

Following the confirmation of panel reliability and normality, all data were subjected to an analysis of variance (ANOVA) using the GLM (general linear model) procedure of SAS statistical software (Version 9.2, SAS Institute Inc., Cary, NC, USA) according to the model for the experimental design indicated by the following equation:*y_ijk_ = µ + s_j_ + a_k_ + sa_jk_ + ε_ijk_*(2)

The terms within the model are defined as: the response (*y_ijk_*) obtained for the *i*th observation for the *j*th sex and *k*th age class, the overall mean (*μ*), the sex main effect (*s_j_*), the age class main effect (*a_k_*), the sex by age interaction effect (*sa_jk_*), and the random error (*ε_ijk_*) associated with response on the *i*th observation in the *j*th sex and the *k*th age class. For post-hoc testing, Fisher’s LSD was used when the main effects/interactions analyzed were significant. A 5% level of significance was used as a guideline to explain significant differences. Pearson’s correlation was used to determine the correlation coefficients for the sensory, physical, and chemical data. Principal component analyses (PCA), using the correlation matrix, and discriminate analysis (DA) were performed to determine and illustrate relationships between the sensory, physical, and chemical data and its association with the samples.

## 3. Results

The physical measurements of the culled warthogs and chemical composition of the back bacon (LT) is presented in Table 4. There were no interactions between sex and age for the physical and chemical measurements. The dead and carcass weight of warthogs differed between sexes (*p* = 0.02 and *p* = 0.05, respectively) and ages (*p* < 0.001 and *p* < 0.001, respectively), where the adult boars were heavier than the adult sows and the adults heavier than the young warthogs, the latter did not differ between sexes, while the whole LTL muscle from adults were heavier (*p* < 0.001) than that of the juvenile warthogs. There was no sex or age effect on the thawing and cooking losses, while the final pick up % was the highest in juvenile females compared to adult males (*p* < 0.01), but the others did not differ from each other. The total weight change % was highest for female adults compared to juvenile males (*p* = 0.03), which also did not differ among the others. There was an interaction on the ash content, with the male juveniles being similar to the female juveniles but having significantly higher ash levels from that of the adults, while that of the juvenile females did not differ from that of the adults (*p* = 0.03). None of the other chemical characteristics of the cooked back bacon differed among the four groups, and were high in protein (~29%) and low in total fat (<2%) contents.

The mean scores for the descriptive sensory attributes are presented in Table 5. During training, attribute descriptors used to describe the aroma and flavor pertaining to boar taint in domestic boars were considered by the panel as aroma and flavor descriptors, but not included in the final statistical analysis as these were not detected by the panelists. The undesirable odor and flavored described as “sour/sweaty” and “fishy” were scored low and there was a significant interaction between sex and age for the fishy aroma (*p* = 0.03), sour/sweaty aroma (*p* < 0.01), and fishy flavor (*p* = 0.05), with all being scored highest in adult males. The residue of the bacon from adult males was higher compared to juvenile females (*p* = 0.04), while the muscle fibers of the bacon from adult females appeared coarser compared to juvenile males (*p* = 0.03), but these attributes did not differ for the others. The other attributes did not differ according to sex or age (*p* > 0.05), with the highest scored attributes including typical bacon and smoky aroma and flavor, and salty flavor, as well as the textural (mouth-feel) attributes tenderness and juiciness.

The fatty acid (FA) compositions of the four groups are presented in Table 6. There was significant interaction between sex and age for the polyunsaturated to saturated (PUFA:SFA) ratio (*p* = 0.01), with the ratio highest for juvenile males compared to adult males and juvenile females but not adult females. The other fatty acids were similar across sex and age (*p* > 0.05). SFA and PUFA contributed primarily to the FA composition, with palmitic (C16:0), stearic (C18:0), linoleic (C18:2ω6), oleic (C18:1ω9c), and arachidonic (C20:4ω6) being the dominant FA. A principle component analysis (PCA) bi-plot (Figure 2) shows the sensory profile for each animal according to physical attributes, proximate composition and FA composition, with Factor 1 explaining 34.59% of the variance. A complete correlation matrix is given in Table 7 to show the significant correlations between sensory attributes and proximate and FA composition. Total pick up % was positively correlated with muscle weight (r = 0.45, *p* < 0.01) and salty flavor (r = 0.60, *p* < 0.01), and shear force was negatively correlated with tenderness (r = −0.60, *p* < 0.01). The panel did not detect any gamey aroma or flavor between the treatments, and therefore no results/correlations were calculated.

## 4. Discussion

The sensory profile of warthog back bacon was dominated by typical bacon and smoky aromas and flavors, and salty flavor. The sensory profile (LL; Table 5) and FA composition (Table 6) of the LT muscles were influenced by the process of bacon production, which are different to that determined for whole cooked warthog LL muscle [42]. To summarize, the bacon had a sensory profile that compares to that of pork bacon regarding aroma and flavor, while tenderness, juiciness, and shear force values did not differ between sex or age class, although the residue and appearance of fibers did. Similar to the findings on cooked warthog muscles [42], there were no differences in the chemical composition between sex and age for the back bacon after cooking, with the moisture content higher, and the protein and fat content lower compared to back bacon produced from South African indigenous pig breeds [4]. The fatty acid composition of the bacon also appears to be influenced by processing, with an increase in the proportion of MUFA and a decrease in the proportion of SFA and PUFA (Table 5). The higher MUFA content is attributed to the higher content of oleic acid (C18:1n9t), while it is noted that elaidic acid was not found in this analysis (Table 6), which were present in cooked warthog LL muscle [42] albeit at very low levels (0.03 mg/g). It is suggested that elaidic acid was not detected as it was present at levels lower than the detection threshold of the technique used. While fat content and FA composition primarily determines the volatiles produced in fresh meat during cooking, the aromatic volatiles develop from lipid degradation and the Maillard reaction, and from interaction between the processes, producing a number of aromatic volatiles that may contribute to the flavor profile [43]. Also, the addition of nitrite in cured meat products is considered to influence the aromatic volatiles through suppression of lipid oxidation and reactivity with FAs during cooking [44].

Since the meat of warthogs have been associated with sensory attributes described as “sour/sweaty” or “fishy” [26], this study provides evidence that processes such as curing and smoking can be used to reduce or “mask” undesirable flavors in meat, thereby converting the meat into a desirable product. The meat from wild animals has been reported to be associated with a gamey or livery taste, while game meat and the meat from non-ruminants is typically high in total unsaturated FA (UFA) content and thus more susceptible to lipid oxidation [15], which leads to undesirable aromas and flavors [16]. However, gamey flavor was not associated with cured smoked salami made from different game meats, which suggests that smoking reduces perceived game flavor in processed products [21,45]. A cured, smoked sausage known as cabanossi in South Africa was produced with warthog meat and did not affect consumer preference compared to the same sausage produced with commercial pork [25]. In this study, it is suggested that the addition of nitrite greatly affect the lipid oxidation during cooking, which determines the volatile compounds produced responsible for flavor. The intramuscular fat (IMF) content is generally low in the meat of game species [16], which primarily consists of structural lipid components, phospholipids, and cholesterol, with high proportions of PUFA. The meat from free-ranging grazers have a higher PUFA, and n-3 in particular, content compared to farmed animals fed concentrate diets, as animals extensively feeding on grasses incorporate more n-3 and n-6 PUFA fatty acids in their muscles [1]. As PUFA oxidize more readily than SFA, meat high in total PUFA is more susceptible to lipid oxidation and the development of associated off-aromas and flavors which decreases meat quality and desirability [15]. As mentioned, processes such as curing, smoking, and addition of spices are used to inhibit the rate of oxidation and subsequent development of off-aromas and flavors, as well as imparting specific flavors [20]. Nitrite as an anti-oxidant stabilizes the heme-iron group of the myoglobin molecule, chelates metal ions and radicals and reacts with UFA [46], while certain phenolic compounds produced from wood-smoking scavenge oxygen radicals [47].

Hydrocarbons, aldehydes, ketones, and alcohols derived from thermal degradation and Maillard reactions are primarily associated with pork flavor, and to a lesser extent, pyrazines, furans, and pyridines [44], as the oxidation of unsaturated fatty acids produce significant quantities of carbonyl compounds (ketones and aldehydes) [48]. Similar aromatic compounds are found in uncured and cured pork but at much lower concentrations in cured pork which help explain the difference in flavor profiles [43]. These lower concentrations [43] are attributed to the suppression of lipid oxidation by nitrite, while certain compounds including pyrazines, pyridines, furans, and nitriles were only present or present at higher concentrations in cooked cured pork [44]. These organic compounds are suggested to contribute to the characteristic flavor of bacon, or in combination with other compounds, which are produced through the interactions between and reactivity of nitrite and FA [43,44]. Pyrazines are the major products of the Maillard reaction and it has been suggested that phospholipids significantly participate in the Maillard reaction which produces heterocyclic compounds [43]. Since phospholipids consist primarily of PUFA, this could explain the desirable bacon flavor profile lean meats such as warthog obtain following processing.

Cured flavor, described here as typical bacon flavor, is the most important characteristic of nitrite-cured meat products, although the dynamics of the flavor development is not fully understood [49]. This flavor is considered a preservation of the fresh meat flavor combined with the retardation of rancidity development in salted meat products, as salting meat accelerates proteolysis and lipolysis which causes the development of rancid aromas and flavors. However, salting also imparts a desirable salty flavor. In addition to nitrite, smoking also reduces lipid oxidation and microbial spoilage [49]. The majority of volatile compounds are derived from smoking and responsible for smoky flavor [50], which cause the meat to have a very different aromatic profile compared to fresh pork or hams [51]. While phenols primarily contribute to smoky flavor, aldehydes, ketones, and alcohols have also been implicated in smoky flavor [52]. The low scores for the fishy and sour/sweaty aroma and flavor of warthog bacon are ascribed to the processing methods used, and these attributes do not contribute strongly to the overall sensory profile of warthog back bacon.

In addition, it is known that meat high in PUFA is undesirable for bacon production (and storage) as the hardness of fat is reduced, which causes difficulties during slicing of bacon as uniform rashers [49]. While the PUFA ratio of warthog meat [53] and bacon is higher compared to domestic pigs, the actual content (mg/g) is very low (Table 6) indicating that the low PUFA content might have a negligible impact on slicing quality. Earlier reports [54] found the adipose tissue of warthogs to consist primarily of palmitic, stearic (SFA, 31%), oleic (MUFA, 20%), and linoleic and linolenic (PUFA, 34%) acids. The warthogs from this investigation had a minimum of subcutaneous fat (data not given), although earlier observations [25,29] has shown that warthogs can develop a significant subcutaneous fat cover depending on season and food availability. Although the total content and composition of the FA in the subcutaneous fat depot is not known, it is suggested that if present the subcutaneous fat layer be removed prior to processing as the fat may contribute to rancidity and slicing quality of warthog back bacon.

The chemical composition of smoked meat products depends on the smoking method used; in reindeer meat, the curing and smoking as preservation process influenced the total fat content to a greater extent than it did the FA composition [55]. The fat content decreased following lipolysis at smoking chamber temperatures of 80 °C and internal meat core temperatures of 65 °C. The weight losses following smoking averaged around 8.5% but there was only a slight reduction in fat content in warthog back bacon, which is expected considering the overall low fat content of warthog meat [25,29] and the low smoking temperatures used (Figure 1). Game meat, characterized by a lean profile, therefore lends itself well as an alternative meat for producing reduced fat processed products; as example, warthog cabanossi was a reduced fat product compared to pork cabanossi although both had similar sensory attributes [25]. Not all products are suited for fat reduction strategies, as fat-reduced products should still satisfy consumers’ expectations regarding distinctive visual qualities, display a level of familiarity, and with the added benefit of being healthier and organic [56], while not compromising eating quality, safety, and production costs [57].

The texture and liking of bacon is influenced by the method of preparation and cooking when applicable, and therefore researchers typically do not attempt to evaluate the liking of texture as “doneness” of bacon is a personal preference [58]. Also, the differences in sample preparation and description of sensory attributes complicates the comparison between the results of studies [16]. Here, the cooking method and presentation to panelists was different compared to other studies on bacon in order to standardize sensory analysis. The moist cooking of bacon as a whole muscle and presentation to the panelists as a cube of meat is suggested to allow for the perception of juiciness, tenderness and appearance of bacon. Despite the different scores for appearance of muscle fiber bundles between sex and age, the scores for juiciness and tenderness was high for warthog bacon and did not differ between sexes or age class. It is noted that the total pick up % and smoking loss % was lower and higher, respectively, for the warthog bacon compared to commercially produced bacon where the desired average pick up is 20% and smoking loss <5%. The differences in the total pick up %, total weight change %, total ash content, and appearance is ascribed to differences in the size (weight) and myofiber composition of the muscles between sexes and age class, while processing is ultimately considered responsible for the sensory profile of warthog back bacon. It is suggested that the high moisture content and low cooking loss of warthog back bacon contributed to the higher initial and sustained juiciness scores.

During mastication, the total moisture contributes to initial juiciness, while fats stimulate saliva secretion and total fat content therefore contributes to sustained juiciness [16]. However, the desired flavor compounds produced during the Malliard reactions further stimulates saliva secretion which could contribute to perceived juiciness [16]. The desired compounds in bacon are associated with attributes described as “bacon,” “fried meat,” “roast meat,” and “cooked meat-like” [44].

A PUFA:SFA ratio of ≥0.45 and omega 6:omega 3 (ω6:ω3) of ≤4 is recommended for the meats consumed by humans as a diet high in unsaturated FA provides health benefits. Warthog back bacon has an improved PUFA:SFA and omega 6:omega 3 ratio compared to South African pig and ostrich bacon, and other smoked meat products (Table 8). Although there were differences between sex and age class, all of the ratios fell within the desired range. Warthog back bacon is also an example of how processed meat products can be developed that meets global health recommendations in terms of low total fat and SFA content. However, the final salt and nitrite content of the bacon was not determined in this study, and future research should determine this as it could have implications for product production, marketing, and labelling. Apart from the recommendation for decreased fat and SFA consumption, there has been concern raised regarding the consumption of red meat and processed product consumption, with calls made for lower salt and nitrite content in processed products.

## 5. Conclusions

The results from this study indicate that warthog meat can be utilized in bacon production with health benefits without compromising the sensory attributes associated with traditional porcine bacon. The study provides evidence that processes such as curing and smoking can reduce or “mask” the undesirable flavors in meat, thereby converting the meat into a desirable product. The addition of nitrite and contribution of smoking compounds are suggested as being responsible for this conversion, and the development of desirable aromatic compounds associated with bacon. It is suggested that the production of processed game meat products should consider physical parameters including age and gender as these affect the size and structure of muscles which could influence the production yields, although this did not appear to ultimately influence the sensory profile of warthog back bacon. Increased utilization of warthog meat has been proposed as a strategy to encourage warthog control and population management, and the utilization of the meat in processed products could broaden the scope of wildlife utilization and game meat consumption.

## Figures and Tables

**Figure 1 foods-09-00641-f001:**
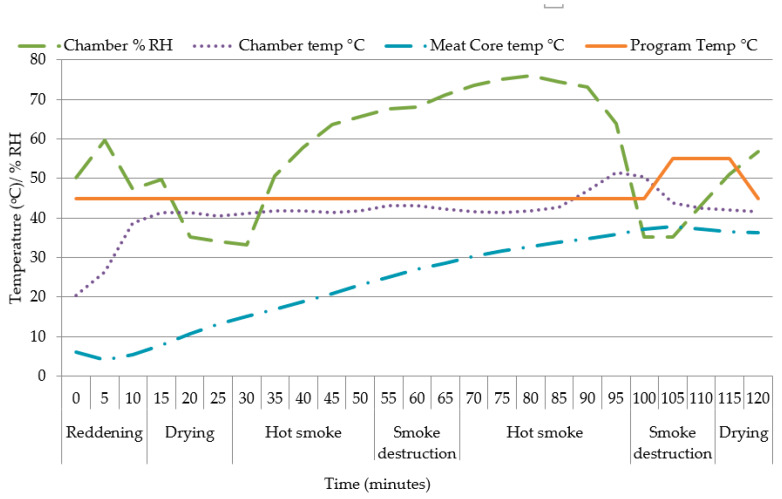
The temperature (°C) and relative humidity (% RH) of the smoking program and temperature (°C) of the chamber and core of *Longissimus thoracis et lumborum* (back) bacon.

**Figure 2 foods-09-00641-f002:**
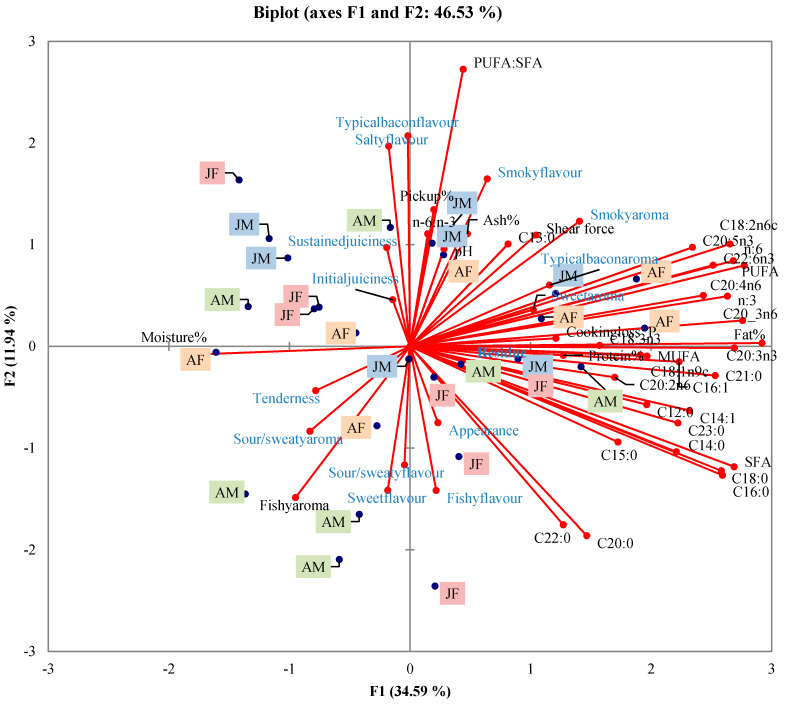
The principal component analyses (PCA) bi-plot of sensory attributes, physical attributes, proximate composition and fatty acid composition of warthog back bacon according to sex and age (AF = adult female, AM = adult male, JF = juvenile female, JM = juvenile male).

**Table 1 foods-09-00641-t001:** Ingredients of the brine (Ref: 01124015) used (Deli SpicesTM).

Component	Ingredients
Seasoning	Emulsifiers (E451, E452, E339), Dextrose, Sugar, Maltodextrin, Sodium Erythorbate (E316), Acidity regulator (Citric Acid [E330]) Ascorbic acid (E300), Anticaking agent (Silicon Dioxide [E551])
Curing salt	Sodium Nitrite [E250], Sodium Nitrate [E251]), Colourant (C.I.45430).
Salt	NaCl

**Table 2 foods-09-00641-t002:** The smoking program of the Reich Airmaster^®^ as used for preparing warthog *Longissimus thoracis et lumborum* (back) bacon.

Smoking Program	Temperature (°C)	Time (min)
Reddening	45	15
Drying	45	15
Hot smoke	45	20
Smoke destruction	45	10
Hot smoke	45	25
Smoke destruction	55	10
Drying	45	10

**Table 3 foods-09-00641-t003:** Definition and scale of each attribute used for the descriptive sensory analysis of warthog back bacon.

Sensory Attribute.	Description	Reference
Aroma ^a^		
Typical bacon aroma	Associated with typical bacon	Back bacon
Smoky aroma	Associated with smoked meat products	Liquid smoke solution
Sweet aroma	Associated with pork loins chops	Pork loin chop
Gamey aroma	Associated with game meat	Fallow deer loin
Fishy aroma	Associated with smoked mackerel	Smoked mackerel
Sour/sweaty aroma	Associated with warthog meat	Warthog fillet
Flavor ^a^		
Typical bacon flavor	Associated with typical bacon	Back bacon
Smoky flavor	Associated with smoked meat products	Back bacon
Salty flavor	Associated with typical bacon	Salt solution
Sweet flavor	Associated with pork loins chops	Sugar solution
Gamey flavor	Associated with game meat	Fallow deer loin
Sour/sweaty flavor	Associated with warthog meat	Warthog fillets
Fishy flavor	Associated with smoked mackerel	Smoked mackerel
Texture (mouth-feel)		
Initial juiciness	Amount of fluid exuded when pressed between thumb and forefinger (0 = Dry, 100 = Juicy)	Chicken breast
Tenderness	Impression formed after first 5 chews using molar teeth (0 = Dry, 100 = Juicy)	Chicken breast
Sustained juiciness	Impression formed after first 5 chews using molar teeth (0 = Tough, 100 = Tender)	Chicken breast
Residue	Amount of residue left in mouth after 10 chews using molar teeth (0 = None, 100 = Abundant)	Pork loin chops ^b^
Appearance of muscle fiber bundles	Appearance of muscle fibers 0 = Fine, 100 = Coarse	Chicken breast

^a^ Scale for descriptors: 0 = Low, 100 = High, unless otherwise stated. Aroma and flavor were analyzed orthonasally and retronasally, respectively; ^b^ Cooked to internal temperature of 75 °C.

**Table 4 foods-09-00641-t004:** The average values (±SD) of the physical measurements of warthog carcasses and chemical composition of back bacon (*Longissimus thoracis)* according to sex and age.

Component	Adult Females (*n* = 8)	Adult Males (*n* = 8)	Juvenile Females (*n* = 8)	Juvenile Males (*n* = 7)	*p* > F		
					Sex	Age	Sex × Age
Body weight (kg)	52.8 ^b^ ± 7.80	63.6 ^a^ ± 9.94	33.0 ^c^ ± 4.96	37.4 ^c^ ± 11.72	0.02	0.02	0.33
Hot carcass weight (kg)	26.8 ^b^ ± 3.49	32.7 ^a^ ± 6.50	17.4 ^c^ ± 4.12	18.0 ^c^ ± 6.25	0.05	0.01	0.17
Cold carcass weight (kg)	26.1 ^b^ ± 3.43	31.871 ^a^ ± 6.44	16.2 ^c^ ± 2.89	17.5 ^c^ ± 6.16	0.01	0.01	0.23
Dress out (%) *	49.6 ^a^ ± 1.67	49.8 ^a^ ± 2.88	49.0 ^a^ ± 2.63	46.4 ^b^ ± 3.26	0.03	0.53	0.91
LTL weight (kg)	0.9 ^a^ ± 0.13	1.07 ^a^ ± 0.24	0.6 ^b^ ± 0.12	0.6 ^b^ ± 0.16	0.55	0.01	0.22
pH_45_	6.2 ± 0.37	6.37 ± 0.23	6.3 ± 0.27	6.2 ± 0.20	0.68	0.98	0.21
pH_24_	5.6 ± 0.01	5.6 ± 0.07	5.6 ± 0.07	5.6 ± 0.09	0.58	0.55	0.54
Thaw loss ^γ,#^ (%)	11.9 ± 2.59	11.840 ± 3.32	11.7 ± 2.47	11.5 ± 2.35	0.79	0.89	0.96
Final pick up ^γ,#^ (%)	14.1 ^a,b^ ± 1.07	13.3 ^b^ ± 1.61	14.7 ^a^ ± 1.00	14.4 ^a,b^ ± 1.48	0.01	0.25	0.29
Smoking loss ^γ,#^ (%)	7.9 ^b^ ± 1.34	7.7 ^b^ ± 1.27	9.1 ^a^ ± 1.75	9.0 ^a^ ± 2.42	0.03	0.53	0.91
Total weight change ^γ,#^ (%)	3.0 ^a^ ± 1.54	2.5 ^a,b^ ± 1.06	1.5 ^a,b^ ± 1.63	1.2 ^b^ ± 2.46	0.01	0.55	0.03
Cooking loss ^#^ (%)	11.3 ± 5.51	12.7 ± 5.40	12.2 ± 4.53	11.8 ± 4.16	0.99	0.77	0.60
Shear force ^#^ (N)	28.1 ± 6.90	27.0 ± 6.03	26.8 ± 2.29	26.0 ± 5.36	0.56	0.62	0.95
Moisture ^#^ (%)	67.7 ± 2.83	67.5 ± 1.30	67.0 ± 1.62	66.6 ± 1.58	0.27	0.68	0.82
Protein ^#^ (%)	29.4 ± 2.65	29.2 ± 1.25	29.6 ± 1.53	29.9 ± 1.61	0.52	0.96	0.75
Fat ^#^ (%)	1.4 ± 0.60	1.3 ± 0.50	1.3 ± 0.42	1.3 ± 0.39	0.74	0.68	0.67
Ash ^#^ (%)	5.5 ^b^ ± 0.75	5.3 ^b^ ± 0.58	5.7 ^a,b^ ± 0.98	6.3 ^a^ ± 0.43	0.03	0.55	0.19

^a–c^ Least square means in the same row with different superscripts are significantly different (*p ≤* 0.05); * Calculated using cold carcass weight; ^γ^ Calculated as % change in weight relative to loin weight; ^#^ Determined for back bacon; pH_45_ = pH measured in loin 45 min post mortem; pH_24_ = pH measured in loin 24 h post mortem.

**Table 5 foods-09-00641-t005:** The mean values (±SD) of the sensory attributes of cooked warthog back bacon according to sex and age.

Sensory Attribute	Adult Female (*n* = 7)	Adult Male (*n* = 7)	Juvenile Female (*n* = 7)	Juvenile Male (*n* = 7)	*p* > F		
					Sex	Age	Age × sex
Typical bacon aroma	72.4 ± 1.98	72.8 ± 0.82	72.2 ± 1.46	72.6 ± 1.00	0.77	0.40	0.99
Smoky aroma	71.9 ± 1.27	71.4 ± 1.46	72.2 ± 1.36	72.4 ± 1.66	0.28	0.76	0.53
Sweet aroma	9.2 ± 0.62	9.2 ± 0.84	9.8 ± 0.35	9.3 ± 0.54	0.14	0.32	0.31
Fishy aroma	1.2 ^b^ ± 0.12	1.5 ^a^ ± 0.24	1.2^b^ ± 0.14	1.2 ^b^ ± 0.12	0.12	0.04	0.03
Sour/sweaty aroma	0.4 ^b^ ± 0.61	0.6 ^a^ ± 0.60	0.3 ^b^ ± 0.58	0.2 ^b^ ± 0.42	0.28	0.88	0.01
Typical bacon flavor	72.4 ± 0.56	72.3 ± 0.81	72.5 ± 0.35	72.9 ± 0.95	0.31	0.62	0.54
Smoky flavor	73.6 ± 0.93	73.1 ± 1.36	72.1 ± 2.37	73.3 ± 1.21	0.27	0.58	0.17
Salty flavor	38.6 ± 2.69	39.6 ± 1.93	39.9 ± 2.40	40.4 ± 1.68	0.22	0.40	0.77
Sweet flavor	4.4 ± 0.93	4.1 ± 1.05	4.5 ± 0.59	3.9 ± 1.47	0.87	0.16	0.50
Sour/sweaty flavor	0	0.7 ± 0.68	0	0	-	-	-
Fishy flavor	3.4 ^b^ ± 1.02	4.1 ^a^ ± 0.86	3.9 ^a,b^ ± 1.41	3.0 ^b^ ± 0.73	0.46	0.72	0.05
Initial juiciness	54.3 ± 1.82	54.4 ± 2.59	53.7 ± 2.69	53.6 ± 2.91	0.50	0.99	0.92
Tenderness	67.9 ± 3.08	67.7 ± 1.98	68.4 ± 2.09	68.8 ± 1.67	0.34	0.91	0.74
Sustained juiciness	66.5 ± 1.55	68.1 ± 2.56	67.6 ± 1.64	68.3 ± 1.27	0.34	0.48	0.11
Residue	22.7 ^a,b^ ± 1.59	23.8 ^a^ ± 0.76	22.1 ^b^ ± 1.05	22.6 ^a,b^ ± 1.11	0.04	0.09	0.53
Appearance of muscle fiber bundles	32.6 ^a^ ± 2.64	31.3 ^ab^ ± 2.15	30.4 ^a,b^ ± 1.78	29.3 ^b^ ± 2.79	0.03	0.20	0.92

^a–b^ Least square means in the same row with different superscripts are significantly different (*p ≤* 0.05).

**Table 6 foods-09-00641-t006:** The average mg/g (±SD) of the fatty acid composition of cooked warthog back bacon according to age and sex.

Fatty Acid	Adult Females	Adult Males	Juvenile Females	Juvenile Males	*p* > F		
					Sex	Age	Sex × Age
Saturated fatty acids							
C12:0	0.05 ± 0.01	0.04 ± 0.02	0.04 ± 0.03	0.03 ± 0.03	0.64	0.62	0.12
C13:0	0.09± 0.04	0.07 ± 0.07	0.05 ± 0.05	0.10 ± 0.04	0.24	0.73	0.25
C14:0	0.07 ± 0.02	0.07 ± 0.04	0.08 ± 0.05	0.07 ± 0.04	0.64	0.68	0.38
C15:0	0.01 ± 0.01	0.01 ± 0.01	0.02 ± 0.01	0.01 ± 0.01	0.29	0.61	0.36
C16:0	2.61 ± 1.01	2.70 ± 1.20	2.78 ± 1.20	2.20 ± 0.81	0.71	0.55	0.40
C18:0	1.66 ± 0.64	1.79 ± 0.66	1.86 ± 0.87	1.45 ± 0.47	0.83	0.61	0.28
C20:0	0.02 ± 0.01	0.02 ± 0.01	0.02 ± 0.01	0.01 ± 0.001	0.82	0.62	0.12
C21:0	0.18 ± 0.07	0.17 ± 0.06	0.16 ± 0.05	0.17 ± 0.07	0.68	0.96	0.68
C22:0	0.01 ± 0.01	0.02 ± 0.01	0.02 ± 0.01	0.01 ± 0.004	0.78	0.80	0.15
C23:0	0.22 ± 0.10	0.23 ± 0.08	0.21 ± 0.01	0.21 ± 0.09	0.73	0.93	0.99
Monounsaturated fatty acids							
C14:1	0.03 ± 0.01	0.03 ± 0.01	0.03 ± 0.01	0.03 ± 0.01	0.78	0.81	0.90
C16:1	0.15 ± 0.06	0.12 ± 0.05	0.12 ± 0.04	0.11 ± 0.05	0.35	0.35	0.25
C18:1ω9c	1.64 ± 0.88	1.23 ± 0.83	1.21 ± 0.62	1.43 ± 0.69	0.65	0.83	0.72
Polyunsaturated fatty acids							
C18:2ω6c	3.82 ± 1.99	2.91 ± 1.47	2.96 ± 0.88	3.53 ± 1.03	0.78	0.70	0.16
C18:3ω3	0.48 ± 0.20	0.43 ± 0.20	0.43 ± 0.20	0.44 ± 0.22	0.58	0.87	0.43
C20:2ω6	0.04 ± 0.02	0.03 ± 0.01	0.03 ± 0.02	0.03 ± 0.01	0.33	0.86	0.31
C20:3ω6	0.38 ± 0.15	0.33 ± 0.12	0.31 ± 0.08	0.34 ± 0.12	0.55	0.76	0.44
C20:3ω3	0.26 ± 0.12	0.26 ± 0.09	0.25 ± 0.09	0.30 ± 0.10	0.66	0.57	0.58
C20:4ω6	1.27 ± 0.49	1.02 ± 0.50	1.03 ± 0.28	1.17 ± 0.42	0.32	0.45	0.09
C20:5ω3	0.38 ± 0.20	0.38 ± 0.16	0.41 ± 0.12	0.51 ± 0.19	0.16	0.46	0.38
C22:6ω3	0.27 ± 0.16	0.22 ± 0.09	0.22 ± 0.06	0.24 ± 0.09	0.65	0.61	0.39
SFA	4.93 ± 1.89	5.11 ± 2.02	5.31 ± 2.20	4.29 ± 1.40	0.78	0.57	0.39
MUFA	1.81 ± 0.94	1.36 ± 0.88	1.70 ± 0.91	1.57 ± 0.74	0.60	0.79	0.68
PUFA	7.28 ± 3.72	5.73 ± 2.59	5.77 ± 1.53	6.85 ± 1.97	0.81	0.77	0.17
PUFA:SFA	1.46 ^a,b^ ± 0.31	1.18 ^b^ ± 0.45	1.20 ^b^ ± 0.36	1.62 ^a^ ± 0.12	0.49	0.64	0.01
ω6:ω3	3.98 ^a^ ± 0.92	3.13 ^b^ ± 0.64	3.14 ^b^ ± 0.71	3.01 ^b^ ± 0.71	0.09	0.08	0.20

^a–b^ Least square means in the same row with different superscripts are significantly different (*p ≤* 0.05); SFA = Saturated fatty acids, MUFA = Monounsaturated fatty acids, PUFA = Polyunsaturated fatty acids, ω6:ω3 = ratio of omega-6 to omega-3 fatty acids.

**Table 7 foods-09-00641-t007:** Correlation matrix between the sensory attributes, proximate composition and fatty acid composition of warthog back bacon.

Variables *	Typical Bacon Aroma	Smoky Aroma	Sweet Aroma	Fishy Aroma	Sour/Sweaty Aroma	Appearance	Initial Juiciness	Typical Bacon Flavour	Smoky Flavour	Salty Flavour	Sweet Flavour	Sour/Sweaty Flavour	Fishy Flavour	Tenderness	Sustained Juiciness	Residue
C12:0	0.221	0.209	0.357	−0.140	−0.193	−0.062	−0.066	−0.152	0.023	−0.209	−0.158	−0.048	0.005	−0.007	0.133	−0.091
C13:0	−0.070	0.122	0.300	−0.129	0.120	0.082	0.224	0.100	−0.026	0.241	−0.221	−0.252	−0.191	0.134	0.163	0.050
C14:0	0.042	0.175	0.290	−0.152	−0.232	−0.048	−0.331	−0.247	−0.001	−0.165	0.165	−0.029	0.259	−0.077	−0.201	0.079
C15:0	0.220	0.242	0.285	−0.311	−0.144	−0.018	−0.263	−0.206	−0.080	−0.228	0.073	0.060	0.127	−0.016	−0.114	−0.047
C16:0	0.251	0.229	0.235	−0.182	−0.169	0.186	−0.157	−0.289	−0.064	−0.308	0.233	0.113	0.187	−0.178	−0.254	0.176
C18:0	0.276	0.222	0.238	−0.077	−0.119	0.097	−0.103	−0.194	0.007	−0.260	0.196	0.126	0.263	−0.222	−0.240	0.250
C20:0	−0.003	−0.048	0.246	0.155	0.165	0.025	0.075	−0.261	−0.300	−0.284	0.128	0.143	0.220	−0.087	−0.217	0.049
C21:0	0.324	0.353	0.218	−0.122	−0.149	0.027	0.060	0.000	0.135	−0.168	−0.127	0.056	0.017	−0.254	0.080	0.066
C22:0	−0.038	0.020	0.091	0.277	0.175	0.136	0.015	−0.271	−0.303	−0.307	0.130	0.109	0.405	−0.030	−0.198	0.050
C23:0	0.322	0.308	0.169	−0.020	−0.079	0.015	0.050	−0.069	0.056	−0.248	−0.104	0.135	0.046	−0.247	0.010	0.005
C14:1	0.334	0.341	0.272	−0.054	−0.123	−0.029	0.004	−0.011	0.114	−0.173	−0.133	0.122	0.058	−0.306	0.011	0.015
C16:1	0.066	0.160	0.162	−0.275	−0.176	0.172	−0.068	−0.191	0.093	−0.142	0.167	−0.102	0.143	−0.218	−0.124	0.261
C18:1ω9c	0.149	0.247	0.174	−0.184	−0.232	0.200	−0.140	−0.164	0.128	0.042	0.180	0.030	0.205	−0.149	−0.319	0.336
C18:2ω6c	0.309	0.470	0.245	−0.422	−0.294	0.052	0.013	0.157	0.329	0.110	−0.101	−0.095	−0.068	−0.248	0.043	0.260
C18:3ω3	0.191	0.356	0.245	−0.139	−0.186	0.014	−0.277	−0.150	0.112	0.071	−0.019	−0.013	0.147	0.095	−0.233	−0.009
C20:2ω6	0.359	0.347	−0.047	−0.098	0.146	0.188	0.026	0.122	0.355	−0.309	−0.159	0.089	−0.090	−0.162	−0.099	0.149
C20:3ω6	0389	0.409	0.227	−0.295	−0.228	0.073	0.066	0.091	0.230	−0.070	−0.108	−0.023	−0.061	−0.272	0.086	0.163
C20:3ω3	0.460	0.566	0.306	−0.190	−0.233	−0.028	−0.094	0.044	0.256	−0.066	−0.188	0.022	0.011	−0.156	−0.032	0.048
C20:4ω6	0.197	0.275	0.047	−0.379	−0.286	0.177	0.020	0.059	0.208	−0.043	0.088	−0.018	−0.133	−0.320	0.054	0.289
C20:5ω3	0.387	0.453	0.349	−0.336	−0.246	−0.104	0.141	0.256	0.277	0.106	−0.333	−0.146	−0.131	−0.139	0.277	0.045
C22:6ω3	0.366	0.402	0.268	−0.338	−0.255	−0.037	0.090	0.215	0.277	0.153	−0.291	−0.071	−0.099	−0.222	0.195	0.113
SFA	0.267	0.248	0.259	−0.145	−0.147	0.142	−0.116	−0.238	−0.029	−0.280	0.179	0.104	0.198	−0.195	−0.217	0.190
MUFA	0.149	0.247	0.177	−0.191	−0.231	0.199	−0.136	−0.166	0.128	0.030	0.179	0.024	0.204	−0.157	−0.310	0.334
PUFA	0.325	0.466	0.239	−0.403	−0.303	0.066	−0.011	0.116	0.300	0.068	−0.088	−0.073	−0.066	−0.238	0.037	0.228
PUFA:SFA	0.175	0.410	0.092	−0.481	−0.311	−0.158	0.085	0.502	0.436	0.492	−0.394	−0.319	−0.399	−0.121	0.303	0.021
ω6	0.292	0.425	0.193	−0.415	−0.295	0.093	0.019	0.130	0.300	0.056	−0.050	−0.071	−0.089	−0.278	0.049	0.271
ω3	0.387	0.533	0.366	−0.292	−0.280	−0.040	−0.114	0.046	0.254	0.098	−0.210	−0.070	0.026	−0.055	−0.012	0.038
ω6: ω3	−0.180	−0.100	−0.218	−0.333	0.001	0.150	0.226	0.209	0.069	0.060	0.148	−0.132	−0.198	−0.361	0.132	0.323
Moisture %	−0.105	−0.084	−0.237	0.260	0.251	−0.021	−0.107	−0.065	0.083	−0.108	0.132	0.063	−0.244	0.062	−0.232	0.005
Protein %	0.068	0.016	0.205	−0.198	−0.179	0.103	0.166	0.015	−0.153	0.017	−0.081	−0.114	0.210	−0.071	0.274	−0.046
Fat %	0.310	0.404	0.263	−0.321	−0.271	0.126	−0.073	−0.047	0.189	−0.052	0.045	−0.002	0.069	−0.236	−0.113	0.262
Ash %	−0.116	0.036	−0.082	−0.288	−0.373	−0.353	0.058	0.242	0.108	0.422	−0.270	−0.115	−0.057	0.030	0.233	−0.236
Shear force (N)	−0.226	−0.526	−0.568	0.510	0.429	0.383	0.250	−0.363	−0.207	−0.378	0.212	0.525	0.223	0.040	0.000	0.379

* Correlation values in bold within the same row indicate significant differences (*p* < 0.05). Values in bold have a *p*-value of > 0.5; SFA = Saturated fatty acids, MUFA = Monounsaturated fatty acids, PUFA = Polyunsaturated fatty acids, ω6:ω3 = ratio of omega-6 to omega-3 fatty acids.

**Table 8 foods-09-00641-t008:** The fatty acids composition (%) of warthog muscle, warthog back bacon other products.

Sample	SFA (%)	MUFA (%)	PUFA (%)	PUFA:SFA	ω6:ω3	Total Fat (%)
Warthog loin [29]	41.3	2.4	56.3	1.4	2.9	1.2
Warthog back bacon *	36.5	12.4	49.4	1.37	3.32	1.3
Pork back bacon [4]	43.3	47.3	9.4	0.2	8.6	8.4
Smoked reindeer [55]	36.3	34.0	29.4	0.9	5.9	3.3

* This study; SFA = Saturated fatty acids, MUFA = Monounsaturated fatty acids, PUFA = Polyunsaturated fatty acids, ω6:ω3 = ratio of omega-6 to omega-3 fatty acids.

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
