# Peer review of "Profile of Back Bacon Produced From the Common Warthog"

_foods, 2020, doi:10.3390/foods9050641_

Round 1
Reviewer 1 Report
Dear authors,
The paper deals with an interesting but somewhat exotic topic. There has been a tendency to emphasize the healthiness of game meat as well as the opportunities to use them more widely than currently, for human nutrition. We must, however, remember that the share of mammalian production animals is 60% of mammal biomass, and that of humans 36% and wild mammals only 4%, respectively. In addition, when game will be produced more intensively, the body composion has a tendency to turn fattier, and muscles more white in type (that will take much time, though). I do not argue againts the use of game meat, but my comment deals more with the tune by which the opportunities of game are discussed, and in this case, in Introduction and Discussion. I say that thinking the global perspective; locally it is different.
The design and methods as well as execution of this study are really very good indeed. I have only minor comments on the paper.
L21. ’here’ --> There
L29. Please consider ’high production alternative’ in the light of the general comment above.
L68. Cabanossi, later always cabanossi.
L107-108. I do not fully understand what did you do with the other parts than the right LL. See the the other parts of the Materials and Methods. Or should it be LTL?
L133. ’final’. Should it be something like ’another’?
L116-135. I am not familiar with the bacon product that you have. The muscle in very lean. There was 1.5 kg mixture added to 10 kg water (or, was the curing salt another addition, and then, how much?). What was the meat/brine relation? I would have interpreted the size differences the other way around and followed the industrial practices. Would it have been better to inject the muscles in order to reduce the effects of size differences, and then put them into a cover brine. It is so that when salt solution is added, it is the meat water that first starts to move outwards, and then later there will be an equilibrium, unless there will be a stable salt gradient within the product created (water activity the same, but salt/water relationships vary).
The salt content is very low in the brine (or was it?) and then even lower in meat. The phosphates that are for some reason called ”emusifiers” (which they are not) may not be very effective in a low salt content. The last one of them, monophosphate, is very inefficient in improving water-holding, but may have influenced on pH. I also wonder the use of nitrate there. It could have an effect if reduced by bacteria to nitrite, but the reaction is not very effective in so low temperature. It is odd to have ascorbic acid and nitrite in the same solution for a longer period of time, as they react to form gaseous nitric oxide (NO).
The product is called ”cooked”. This does not fulfill the definiton of cooking to me, as the core temperature was below 40 C and the surface about 50 C within 145 minutes. From the safety point of view it is something between short term high temperature incubation and very low temperature cooking. New type of product to me.
What does 'smoke destruction' mean?
Figure 1. The time scale of the cooking process is not very clear and seems not to be in perfect concert with the scheduled program. See e.g. that the chamber temperature is higher than the set value and then clearly lower than that, close to the end.
L140. Please explain T’s. How did you do this, as the muscles were excised and cooked? Maybe it was an estimate, roughly?
Table 4. I do not recommend to show two decimals for most of the parameters, as sometimes the SD is about 50% of the mean, i.e. it means that the second decimals do not make sense as also the integrals of the mean are not ”sure”. This has also a bearing to the not-so-expected results that cooking loss is negatively correlated with moisture (could be logical also!) and positively with protein content. Moisture was negatively correlated with protien. L262-265. I would be careful in concluding too much about these aspects, when the variations are so large and the effects of confounding factors were not excluded (?, like pH, size of the muscles). I wonder also, how much there is seasonal and regional variation.
Table 5. Again, consider reducing the number of decimals into one.
Table 6. You have used much time and effort for determining the results. The sum of fatty acids is, however, barely over 1%. Because the whole carcass will be used, would it have been better to determine the carcass fats as well to obtain information about the nutritional relevance of the animal of which the bacon is only a minor part? I admit though that the paper dealt with bacon and that the correlation matrix provides interesting material about the significance of different fatty acids, provided that they are free fatty acids in significant quantities.
Discussion. It is sometimes difficult to follow when your discussion deals with your own results and when cited results, as past tense is used throughout and sometimes with citations and sometimes without, and the sometimes the citation covers only a technical aspect in the sentence. Example; L398-401.
L320-323. Where are the results of uncured LL? Table 8? Is Table the relevant citing here?
L324-325. Where the increase is shown? Not in Table 5.
L366. What are ’organic nitrite combounds’?
L378. 'nitrites' --> nitrite.
L385-388. Low and high PUFA. The message is mixed. On the first hand high PUFA is bad for slicing, because the fat is oily. But here the message has been that the PUFA in fat is high, but naturally in this lean meat it low within the total content of the meat. Please find another way to express this so that you would not claim that warthog is low in PUFA.
L423-425. Are you sure that that these two aspects are causatively linked (intial moisture (added water included?) content --> juiciness; and cooking loss associated to the moisture content. There is a possibility that the negative correlation is not a relevant finding, but I believe that the water content after cooking is (water-holding).
L425-428. The message in this is rather unclear and does not sound right. Therefore I checked the reference, and it does not seem to be the right one either.
Conclusions. The conclusions are rather general and paint the opportunities of warthog bacon with rather bright colors. I just wonder are they too far-reaching when considering the production possibilities and the fact that there were no standard product for comparisons, using the same technology? (In training of the paper there were reference samples from normal pork?) I do not, however, suggest changes.
Technical. Why there are two different years for the issue (2019, 8 and 2020, 9)?
Author Response
Reviewer I Warthog bacon
The paper deals with an interesting but somewhat exotic topic. There has been a tendency to emphasize the healthiness of game meat as well as the opportunities to use them more widely than currently, for human nutrition. We must, however, remember that the share of mammalian production animals is 60% of mammal biomass, and that of humans 36% and wild mammals only 4%, respectively. Although I agree with this comment, I am also of the opinion that the amount of red meat (of which some is derived from wild animal/bushmeat) consumed in the informal market is totally under-estimated. Most of our data on meat production/consumption is derived from the formal market where there are food safety inspection, etc and it is thus easier to keep record of the volumes. In addition, when game will be produced more intensively, the body composion has a tendency to turn fattier this I think is more applicable when additional feed is fed to the animals, and muscles more white in type (that will take much time, though) agreed that this will take and also will be more applicable when the animal’s activities is curtailed by e.g. keeping them in smaller pens/paddocks. I do not argue againts the use of game meat, but my comment deals more with the tune by which the opportunities of game are discussed, and in this case, in Introduction and Discussion. I say that thinking the global perspective; locally it is different.
The design and methods as well as execution of this study are really very good indeed. I have only minor comments on the paper.
L21. ’here’ --> There Changed
L29. Please consider ’high production alternative’ in the light of the general comment above. ‘high production’ deleted to fit into the thoughts/comments above.
L68. Cabanossi, later always cabanossi. Changed to lower capitals “cabanossi”
L107-108. I do not fully understand what did you do with the other parts than the right LL. See the the other parts of the Materials and Methods. Or should it be LTL? Thank-you for indicating this ambiguity, the sentences have now been changed and we hope that the meaning is now clearer.
L133. ’final’. Should it be something like ’another’? we have removed the term “ change” so that we can get a better idea of the meaning.
L116-135. I am not familiar with the bacon product that you have. The muscle in very lean. There was 1.5 kg mixture added to 10 kg water (or, was the curing salt another addition, and then, how much?). The 1.5 kg mixture also contained the curing salt and salt; we have rephrased this section and hope that it is now easier understood.
What was the meat/brine relation? I would have interpreted the size differences the other way around and followed the industrial practices. In South Africa, it is typical to inject the back-bacon and then leave the injected samples in the same brine overnight in a chiller before smoking the next day. Would it have been better to inject the muscles in order to reduce the effects of size differences, and then put them into a cover brine. We did evaluate this option but decided against this as some of the eye muscles were extremely small and we were of the opinion that an overnight soak would have ensured that equilibrium in the movement of salt/water as described below by the reviewer, would have been obtained. Also, your small scale processors who would typical make warthog bacon would not have an injector and would therefore soak the product. Where applicable, we have change the nuance of the paper to reflect more that this would be a typical “artisan” product. It is so that when salt solution is added, it is the meat water that first starts to move outwards, and then later there will be an equilibrium, unless there will be a stable salt gradient within the product created (water activity the same, but salt/water relationships vary).
The salt content is very low in the brine (or was it?) and then even lower in meat. This is typical in South Africa where most manufacturers have moved to produce “low” salt bacon. What is also interesting is that South Africa still uses nitrite (as nitrite or as nitrate + acid) to give the classical “pink/red” colour under vacuum packaging conditions. All the bacon in South Africa is sold vacuum packed. It is interesting that in South Africa, we have moved to low salt bacon, but then we retain the use of nitrites! The phosphates that are for some reason called ”emusifiers” (which they are not) may not be very effective in a low salt content. The last one of them, monophosphate, is very inefficient in improving water-holding, but may have influenced on pH. I also wonder the use of nitrate there. It could have an effect if reduced by bacteria to nitrite, but the reaction is not very effective in so low temperature. It is odd to have ascorbic acid and nitrite in the same solution for a longer period of time, as they react to form gaseous nitric oxide (NO). The reviewer is correct in the comments around the science of the various chemicals added. As this was a commercial mixture, we attempted to gain clarity from the manufacturers around the different chemical components but they quoted IP rights and did not provide further information.
The product is called ”cooked”. This does not fulfill the definiton of cooking to me, as the core temperature was below 40 C and the surface about 50 C within 145 minutes. From the safety point of view it is something between short term high temperature incubation and very low temperature cooking. The reviewer is correct; all reference to “cooked” has been removed where applicable. Do note tough, that for the sensory evaluation, the bacon was cooked further before consumption. New type of product to me. You are correct; this is typical of the commercial process in South Africa. It is also the norm in South Africa to cook the bacon further before consumption.
What does 'smoke destruction' mean? This is new technology that has been fitted to this particular smoker with the idea that it condenses and destroys some of the smoke particles to decrease the pollution effect. Also bear in mind that this apparatus uses friction to create smoke and not a smouldering fire as is used in older types of industrial smokers.
Figure 1. The time scale of the cooking process is not very clear and seems not to be in perfect concert with the scheduled program. See e.g. that the chamber temperature is higher than the set value and then clearly lower than that, close to the end. I think the reviewer may have misread the RH in the chamber as the Chamber’s temperature. The only time the latter goes above 45°C is in the “smoke destruction” phase after 85 mins.
L140. Please explain T’s. How did you do this, as the muscles were excised and cooked? Maybe it was an estimate, roughly? T – Thoracic & L = lumbar. From the deboning, we could see where the ribs were on the muscle and thus we were able to give an accurate indication where we split the whole long muscle into the front thoracis and back lumbar regions.
Table 4. I do not recommend to show two decimals for most of the parameters, as sometimes the SD is about 50% of the mean, i.e. it means that the second decimals do not make sense as also the integrals of the mean are not ”sure”. This has also a bearing to the not-so-expected results that cooking loss is negatively correlated with moisture (could be logical also!) and positively with protein content. Moisture was negatively correlated with protien. As suggested, the means have all been rounded to 1 decimal points whilst the SD have been maintained at 2 decimal points (for all the Tables where applicable).
L262-265. I would be careful in concluding too much about these aspects, when the variations are so large and the effects of confounding factors were not excluded (?, like pH, size of the muscles). The reviewer is correct, and as the data was limited and the confounding effects were not taken into consideration, these lines dealing with correlations were deleted from the manuscript. I wonder also, how much there is seasonal and regional variation. Both season and region have an effect as the seasons (summer vs winter) are extreme in the areas where these animal occur and thus towards the end of winter their condition is poor with hardly any fat in their muscles or subcutaneously. As pertaining to region, this effect is very pronounced where there are crops grown (such as potatoes or maize/corn) and the warthogs are actively foraging in the cultured lands. So much so, that they are hunted as pests. The warthogs from this trial were not from a region where any crops were grown and thus they were totally dependant on natural vegetation for their food.
Table 5. Again, consider reducing the number of decimals into one. Has been done as suggested.
Table 6. You have used much time and effort for determining the results. The sum of fatty acids is, however, barely over 1%. Because the whole carcass will be used, would it have been better to determine the carcass fats as well to obtain information about the nutritional relevance of the animal of which the bacon is only a minor part? Although this would have been of value when the carcasses have a substantial subcutaneous fat cover (as is typical for domestic swine), warthog tend to have very little subcutaneous fat cover, particularly in this experiment where there was very little natural food for the animals. Please also note that these fatty acids are given as mg/g bacon and this is the reason why the values are so low. We thought it would be best to give the data in this format as this is the amount of specific f/a a person would be consuming. Due to the low levels of the f/a we have retained the 2 decimal points in this Table.
I admit though that the paper dealt with bacon and that the correlation matrix provides interesting material about the significance of different fatty acids, provided that they are free fatty acids in significant quantities.
Discussion. It is sometimes difficult to follow when your discussion deals with your own results and when cited results, as past tense is used throughout and sometimes with citations and sometimes without, and the sometimes the citation covers only a technical aspect in the sentence. Example; L398-401. We have tried to clear this issue in our revision
L320-323. Where are the results of uncured LL? The chemical composition of the uncured LL were not analysed – in hind sight, a short coming. Table 8? Is Table the relevant citing here? The citing of Table 8 would not be relevant here as the reference (29) is applicable to raw muscle from a different set of warthogs that were from a different region and culled over different seasons.
L324-325. Where the increase is shown? Not in Table 5. These lines have been removed as we were comparing the chemical composition of the cooked bacon to another set of cooked loins from a different group of warthogs – which is actually of little value as there are too many confounding factors that differ between the two groups.
L366. Now L 361. What are ’organic nitrite combounds’? ‘nitrite’ deleted
L378. 'nitrites' --> nitrite. changed
L385-388. Low and high PUFA. The message is mixed. On the first hand high PUFA is bad for slicing, because the fat is oily. But here the message has been that the PUFA in fat is high, but naturally in this lean meat it low within the total content of the meat. Please find another way to express this so that you would not claim that warthog is low in PUFA. Has been rephrased.
L423-425. Are you sure that that these two aspects are causatively linked (intial moisture (added water included?) content --> juiciness; and cooking loss associated to the moisture content. There is a possibility that the negative correlation is not a relevant finding, but I believe that the water content after cooking is (water-holding). Now Lines 418. The reviewer has a valid question and to address this, the section linked to the correlations has been removed.
L425-428. The message in this is rather unclear and does not sound right. Therefore I checked the reference, and it does not seem to be the right one either. After reading the reviewer’s comments and revisiting the sentence, we agree with the reviewer in that the statement does no make sense – so the statement and reference has been removed.
Conclusions. The conclusions are rather general and paint the opportunities of warthog bacon with rather bright colors. I just wonder are they too far-reaching when considering the production possibilities and the fact that there were no standard product for comparisons, using the same technology? (In training of the paper there were reference samples from normal pork?) I do not, however, suggest changes. We have rewritten the Conclusion to “tone down” the value of warthog bacon, after also taking into consideration the opening remarks given by this reviewer.
Technical. Why there are two different years for the issue (2019, 8 and 2020, 9)? This is on the template we used; I assume that the editor will change this if the manuscript is accepted?
Reviewer 2 Report
The manuscript entitled “Profile of Back Bacon Produced From the Common Warthog” documents a study that aims to explore the feasibility of utilizing wild warthog as the protein source for the production of bacon. The methodology of the study is well executed, aside from a few clarification questions. The authors conclude that warthog loin can be utilized to produce back bacon and a lot of the negative flavor notes associated with this meat are eliminated by processing.
The one major issue the authors need to address is the explanation of the results. As currently written, it is unclear when the authors are comparing means because of the interaction. Additionally, the authors do not always state the status of the main effects once they declare the status of the interaction. Therefore, for all dependent variables, the authors need to clearly state the status of the interaction first, then address the main effects if the interaction is not significant. The authors also need to separate main effect means as males vs. females and young vs old, which they do not do in several locations.
Line Specific:
Line 61: Should be "omega 6 to omega 3" not "omega 3 to omega 3".
Lines 126-127: Was the largest, median, or smallest loin temperature measured?
Line 134: Are all RH supposed to be zero in the table? This does not match Figure 1.
Figure 1: Authors need a y-axis label for RH. Reader has no idea what the changes are?
Lines 142-144: In a commercial hog there would be more fat in the LT than the LL? Why weren't both areas measured chemically?
Line 146: Cooking methodology needs to be outlined because the second cooking makes no sense.
Line 165: Where did the references lie on the scales?
Lines 251-255: Data are presented and separated as if there were a significant interaction, but there was not one. The comparisons should be boars vs. sows and adults vs. juveniles.
Line 258: This comparison would imply the interaction is significant but that has not been declared in the text or table. State if interaction is significant. If covered by lines 251-252, this comparison should not be made.
Lines 274-277: It would benefit the manuscript for the authors to explain the interaction in terms of how the responses of males and females within the ages differed.
Lines 277-279: is this a significant interaction the authors are writing about?
Table 5: Why aren't means separated for the significant interaction for "sour/sweaty aroma"? They need to be. The authors need to also have interaction and main effect p-values on this table and all the others.
Lines 287-289: Same comment as other interaction mean separations.
Line 321: Wasn't chemical composition measured on the LT portion of the muscle. The LL was not measured chemically.
Lines 335-338: The authors cannot really state this given they did not conduct sensory panel analysis on uncured pork. Their scale would say it is just low compared to their top anchor.
Author Response
Reviewer 2 Responses
The manuscript entitled “Profile of Back Bacon Produced From the Common Warthog” documents a study that aims to explore the feasibility of utilizing wild warthog as the protein source for the production of bacon. The methodology of the study is well executed, aside from a few clarification questions. The authors conclude that warthog loin can be utilized to produce back bacon and a lot of the negative flavor notes associated with this meat are eliminated by processing.
The one major issue the authors need to address is the explanation of the results. As currently written, it is unclear when the authors are comparing means because of the interaction. Additionally, the authors do not always state the status of the main effects once they declare the status of the interaction. Therefore, for all dependent variables, the authors need to clearly state the status of the interaction first, then address the main effects if the interaction is not significant. The authors also need to separate main effect means as males vs. females and young vs old, which they do not do in several locations. We take note of this comment around the interaction and main effects, and where applicable, we have addressed this issue
Line Specific:
Line 61: Should be "omega 6 to omega 3" not "omega 3 to omega 3". Thank-you for picking up this error, it has now been changed
Lines 126-127: Was the largest, median, or smallest loin temperature measured? It was measured in a medium sized bacon – has been changed in the text.
Line 134: Are all RH supposed to be zero in the table? This does not match Figure 1. Thank-you for showing this, Table 1 is the program used for the smoker. During the smoking, the RH was not set, only measured. This column has been removed from the Table.
Figure 1: Authors need a y-axis label for RH. Reader has no idea what the changes are? Thank-you for noticing this, the y-axis has now been adjusted as has the title of the Figure.
Lines 142-144: In a commercial hog there would be more fat in the LT than the LL? Why weren't both areas measured chemically? This is correct as pertaining to domestic swine, however the warthog subcutaneous fat were so thin that the differences were deemed to be insignificant.
Line 146: Cooking methodology needs to be outlined because the second cooking makes no sense. The wording has been changed as per Reviewer 1’s questions. The smoking was not an actual cooking per se. In South Africa bacon is smoked at a low temperature and then packed and sold. The consumer normally cooks the bacon. Therefore the reason for cooking the bacon before sensory evaluation. Also note that normally the bacon is sliced before pan frying, but in this investigation the whole loin was cooked as it was thought that this method would give a more uniform cooking treatment to the whole bacon loin than frying slices.
Line 165: Where did the references lie on the scales? Now L156 – the reference samples were all lying towards the top of the scale as the training sessions gave an indication of the overall attributes for the warthog bacon.
Lines 251-255: Data are presented and separated as if there were a significant interaction, but there was not one. The comparisons should be boars vs. sows and adults vs. juveniles. Now L241-245. The sentence has been corrected.
Line 258: This comparison would imply the interaction is significant but that has not been declared in the text or table. State if interaction is significant. If covered by lines 251-252, this comparison should not be made. Line 245 – with the correct of the earlier sentence, this sentence is now correct. We first tested for an interaction, and where there was an interaction this was discussed. Where there was no interaction, the main effects (if present) were discussed.
Lines 274-277: It would benefit the manuscript for the authors to explain the interaction in terms of how the responses of males and females within the ages differed. In L252-253, we state: “There were no interactions between sex and age for the physical and chemical measurements.” We then discuss the influence of the main effects that differed significantly, and where there was no main effect, we mention this (rest of the specific paragraph).
Lines 277-279: is this a significant interaction the authors are writing about? This interaction has now been rephrased and described, now in L260-262
Table 5: Why aren't means separated for the significant interaction for "sour/sweaty aroma"? They need to be. These have now been included.
The authors need to also have interaction and main effect p-values on this table and all the others. We have not given the specific P-values in the Table as we are of the opinion that the superscripts with their relative Table footnotes describe these. Where we feel the differences need to be highlighted, we have given the actual p-values in the text.
Lines 287-289: Same comment as other interaction mean separations. These have now been included.
Line 321: Wasn't chemical composition measured on the LT portion of the muscle. The LL was not measured chemically. The reviewer is correct, thank-you for noticing this error.
Lines 335-338: The authors cannot really state this given they did not conduct sensory panel analysis on uncured pork. Their scale would say it is just low compared to their top anchor. These sentences have been rephrased.
Round 2
Reviewer 1 Report
No further comments after the amendments.
Reviewer 2 Report
Thank you to the authors for considering all the edits and comments on the first round of reviews. I am strongly going to suggest the authors reconsider adding main effect and interaction P-values to all tables. Tables and figures as supposed to stand on their own and as data are presented now, these tables do not stand alone. If the in interactions is not significant, the reader has no idea if the main effects are die to lack of P-values.
Author Response
The specific P values for the main effects of Age, Sex and their interaction has now been added into each Table as requested
Round 3
Reviewer 2 Report
Thank you for your consideration.